# The Transition of Cities towards Innovations in Mobility: Searching for a Global Perspective

**DOI:** 10.3390/ijerph19127197

**Published:** 2022-06-11

**Authors:** Cayetano Medina-Molina, María de la Sierra Rey-Tienda, Eva María Suárez-Redondo

**Affiliations:** 1Área Departamental Ciencias Sociales y de la Salud, Centro Universitario San Isidoro, 41092 Sevilla, Spain; 2Facultad de Ciencias Jurídicas y Económicas, Universidad Isabel I, 09003 Burgos, Spain; 3Cátedra Metropol Parasol de Gestión Sostenible y Dinamización Comercial Innovadora de Espacios Singulares en Entornos Urbanos, Universidad de Sevilla, 41004 Sevilla, Spain; sierrarey@gmail.com; 4Dpto. Administración de Empresas y Marketing, Facultad de Ciencias Económicas y Empresariales, Universidad de Sevilla, 41004 Sevilla, Spain; esredondo@us.es

**Keywords:** smart mobility, multi-level perspective, qualitative comparative analysis, cluster analysis, city readiness, sustainable transitions

## Abstract

The growing concentration of the population in urban areas presents great challenges for sustainability. Within this process, mobility emerges as one of the main generators of externalities that hinder the achievement of the Sustainable Development Goals. The transition of cities towards innovations in sustainable mobility requires progress in different dimensions, whose interaction requires research. Likewise, it is necessary to establish whether the experiences developed between cities with different contexts can be extrapolated. Therefore, the purpose of this study was to identify how the conditions that determine a city’s readiness to implement urban mobility innovations could be combined. For this, qualitative comparative analysis was applied to a model developed using the multi-level perspective, analyzing 60 cities from different geographical areas and with a different gross domestic product per capita. The R package Set Methods was used. The explanation of the readiness of cities to implement mobility innovations is different to the explanation of the readiness negation. While readiness is explained by two solutions, in which only regime elements appear, the negation of readiness is explained by five possible solutions, showing the interaction between the landscape and regimen elements and enacting the negation of innovations as a necessary condition. The cluster analysis shows us that the results can be extrapolated between cities with different contexts.

## 1. Introduction

A global increase in the population living in cities has taken place, leading to projections such as 70% living in major urban centers by 2050 [1]. This is a process linked to infrastructures and lifestyles, and that entails externalities for the environment and society with implications for the health of its citizens [2,3,4,5,6]. The urban environment is linked to sustainable development, health and well-being, being considered one of the greatest challenges for environmental health [7,8,9]. Mobility, a complex urban system with multiple interactions, is gaining relevance within the urbanization process and plays a critical role in the daily life of cities [10,11,12]. The modernization of urban mobility plays a key role in the healthiness and sustainability of cities [11,13,14], prompting cities of all sizes to have sustainable mobility goals on their agenda [15]. Any reference to sustainability must bear in mind the Sustainable Development Goals (SDG), through which the United Nations established 17 global goals designed as a plan to achieve a better and more sustainable future for all [16]. Sustainability is one of the three key elements—together with safety and intelligence—of a modern mobility system [12], meaning that sustainable mobility is linked to different SDGs (see Table 1). This is clearly reflected in SDG11, which aims to achieve the development of sustainable, inclusive, safe and resilient smart cities, ensuring socio-economic growth and high living standards, and paying attention to the benefits generated for society, the environment and mobility [17,18].

The greater the concern for the SDGs, the greater the attention should be paid to the events that take place in cities [8]. Such is the relevance of urban transformation for sustainability that many of the SDGs find in the city the necessary scale for its achievement [19], with cities being considered the drivers of sustainable development [17]. For this reason, cities around the world demonstrate the need to transform their urban mobility systems into more sustainable ones [20]. While the traditional approach to the smart city focuses on the provision of services through technologies, a later approach aligned with the SDGs is not only committed to developing a smart city that helps to define and solve social challenges, but also to increasing the well-being of citizens and meet their needs through a flexible and efficient conception that incorporates the human capital variable [5,6,21,22]. In fact, the smart city offers citizens a good environment to live in and increases their quality of life. From the SDG perspective, the smart city emphasizes the importance of providing citizens with health and well-being, as cities cannot be truly smart without being sustainable [6,21]. 

To explain the processes and dynamics manifested in cities, as well as the interactions between cities and their context, urban transformation recognizes radical innovations as determinants in achieving social and environmental improvements [2,19,23,24]. In this sense, the innovations in mobility become a key point, helping to solve many of the obstacles presented by urbanization, and leading to an improvement in the quality of life of inhabitants, favoring a sustainable future [2,21,25,26]. Since the relevance of the implementation of mobility innovations is evident for cities to face mobility challenges, this paper answers the following research question: are there combinations of conditions that explain the readiness of cities for the implementation of innovations in mobility? Therefore, the purpose of this paper is to identify how the conditions that determine a city’s readiness to implement urban mobility innovations could be combined.

A city’s smartness is measured based on the factors that determine its level of readiness for transformation and change [2,27,28]. Therefore, the components that contribute to the creation of smart cities should be carefully analyzed to identify the areas in which work should be undertaken [2]. Little attention has been paid to assessing the necessary level of readiness for smart city transformation [27,28], and few studies have explained the factors that determine its implementation [2,28,29]. Thus, considering the readiness of smart city is an area of study that has rarely been addressed [28]. We therefore identify as the first research gap of this paper the possibility of identifying the combinations of factors that determine the level of readiness of cities to implement sustainable mobility innovations. In this field, several frameworks have been used to determine the readiness of smart cities [30].

The smart city is the result of a transformation process followed by many cities around the world aiming to increase their efficiency, facilitate the participation of their citizens, and reduce the environmental impact of their activities [1]. Although it is an ambitious concept, and many countries prioritize smart cities as a way to strengthen their sustainability strategies [25], it is difficult to identify shared definitions and guidelines across countries with different contexts because the development of smart city projects is tailored to each city [4,6,21]. This is because local governments attribute different meanings to urban innovations and formulate ad hoc place-based smart city strategies related to their contexts, seeking to solve their main problems [6,21]. In this sense, smart mobility has not been established in the same way for cities around the world [26]. Therefore, the second research gap is the investigation of the conditions that determine the implementation of sustainable mobility innovations to different urban contexts [21]. This responds to the need to link perspectives and overcome limitations by linking place-based learning, thus enabling city-to-city learning [19]. It also responds to the need to broaden the focus of research on city transitions to sustainability by developing work beyond Europe [23]. 

Mass urbanization and resource scarcity are global phenomena for which urban transformation towards a smart city is the most viable solution [31]. Sustainable innovation is the result of the combined effect of different elements [32]. In this sense, the building blocks of socio-technical transitions in urban mobility rest on windows of opportunity created by the convergence of factors [33]. However, smart city research has employed methodologies unable to analyze causal relationships between factors acting at different levels [2]. Therefore, this paper uses the multi-level perspective (MLP). Furthermore, research has found different trajectories for cities when embarking on smart city projects [28], making it interesting to consider the option of using qualitative comparative analysis (QCA). QCA makes it possible to apply a combinatorial perspective in which elements of the external and internal environment and organizational characteristics are used [34]. The results obtained allow us to establish the different conditions that determine the readiness of a city to implement mobility innovations, or its negation. Likewise, it is possible to extrapolate the experiences of the cities analyzed between different contexts.

## 2. Materials and Methods

The purpose of this article was to identify how the conditions that determine the level of readiness among cities for implementing urban mobility innovations could be combined. To achieve this, we developed a model supported by the MLP, since it has been used to explain the implementation and adoption of innovative mobility solutions [13,35,36,37]. Next, we explained the justification of the propositions and the choice of data source. Fuzzy set QCA (fsQCA) was employed for data analysis because it is recommended to use QCA when trying to establish the effect of multiple and complex causality, as in the case of MLP [35,38]. Likewise, QCA has been used to explain the complex interconnection of conditions at different levels involved in the diffusion of mobility innovations [35,39,40,41].

### 2.1. Model

The study of environmental transformations acting across different levels commonly employs the MLP of sustainable transitions in socio-technical systems [37,42]. A socio-technical system is a network of actors, institutions and norms linked to technologies [27]. Based on these, a transition is defined as a social process of change in culture, structure and social practices specific to the contexts in which they occur across different socio-technical systems [27,43,44]. The MLP of socio-technical systems enables understanding the co-evolutionary and multi-level interactions of infrastructure, technology and knowledge to create specific regimes. The basic principle of the MLP is that transitions to sustainable systems are non-linear processes resulting from processes taking place at three different analytical levels—landscape, regime and niche—so that the success of an innovation is not based on its intrinsic robustness, but on the interaction of its three analytical levels [33,42]. 

The landscape—the macro level—consists of deep structural trends and social or physical conditions—exogenous environment—that shape systems [24,33,42,43]. This represents the broadest socio-technical context [45,46]. This is the level that is most structured, stable over time and that evolves slowly through prolonged stages that help to destabilize the regime [33]. The regime—the meso level—is a semi-coherent set of dominant socio-technical rules, norms and values. It incorporates the socio-technical structure [45,46]. This level provides guidance for the activities of different stakeholders and ensures the coordination and dynamic stability of the socio-technical configuration [24,47]. It is composed of a set of actors and consolidates the domain of technologies in use and limits the range that can be diffused, representing the stabilization of technological trajectories while preserving the status quo [33,42,43]. The niche—the micro level—is a small, emerging network of actors generating radical innovations that diverge from and destabilize the regime [33,42,43]. It is the level that is least structured and can be considered an innovation space embedded in the local context [24]. At this level, experimentation and radical innovations can be developed [45,46]. Innovations developed in niches compete with those technologies that benefit from the socio-technical systems existing around them [33,42]. This is the level at which sustainable solutions linked to urban developments are tested [44].

This paper proposed a model based on the MLP to study the readiness of cities to undertake transitions towards mobility innovations, using components at three levels: landscape, regimen and niche (Table 2). The first two—landscape and regimen—functioned as conditions and the third—niche—as the outcome. In this way, we explained the different conditions that determine the level of readiness of a city to implement innovations in mobility. For the consideration of the landscape or regime conditions, their definitions in the index used for their measurement were taken as a reference. In this way, innovation was taken as a landscape condition, since in its definition it is not linked to mobility and can be considered exogenous.

Among the landscape conditions is the city’s level of innovation. An urban transition must recognize the relevance of innovation, experimentation and initiatives developed, as well as the processes that are specific to them [48]. Cities try to alter urban mobility systems through innovations [20]. Urban innovation is linked to the adoption of technologies that enable the efficient and effective interconnection of its infrastructures [4]. Furthermore, strengthening innovation is among the factors that determine the readiness level of smart city transformation [27,49]. 

Regime conditions include infrastructure, market attractiveness, system efficiency and social impact. Infrastructure comprises durable assets that define the physical configuration of mobility systems [20]. Infrastructure provides a solution to the problems arising from massive urbanization [21,31] and ultimately determines the health of cities [22]. The reinforcement of infrastructure is essential to determining the level of readiness for smart city transformation [1,27,31,44,49,50]. The attractiveness of a city is linked to its ability to attract a creative population [51], a construct that encompasses a broad spectrum of elements [52]. Thus, attractiveness is linked to economic strengths [51] or to citizens’ perception of quality of life [53]. In any case, city governments try to make their cities as attractive as possible, not only for people to live in but also to facilitate investments [50,53]. The growing awareness of cities regarding their resilience to environmental challenges is causing them to embark in activities to identify new forms of urban efficiency [21,54]. In addition, smart cities can increase economic efficiency to a greater extent than those linked to ecological factors [15,25]. Elements that condition the transformation of cities include social impact [31]. Sustainable mobility plays a key role in achieving a sustainable urban environment, generating benefits for the well-being and public health of cities [12,15].

At the niche level, the readiness of a city to implement mobility innovations was used. The transformative capacity of a city is a comprehensive and holistic framework for identifying the relevant contextual conditions to advance changes towards sustainability [48]. Turning cities into sustainable ones requires a transformative process, which considers the context and needs of the city, the local interest, the quality of life of citizens and the readiness of the city for change [3]. Readiness is a multidimensional construct that reflects the simultaneous presence of political, social, economic and environmental factors [27,31,49].

### 2.2. Justification of Propositions

Socio-technical transition (STT) recognizes that cities are complex, adaptive, dynamic and open systems in constant evolution [23,47]. Sustainable transitions present the city as a socio-technical system that needs to be reconfigured, as guided by the SDGs [23,42,55]. From the STT perspective, an urban transformation implies regime shifts as a consequence of the tension between regime rules and landscape, internal regime frictions and the pressure of niche-generated alternatives [47]. The co-evolution between landscape and regime is an important factor in the evolution of the social system. However, the self-organizing nature of the regime only enables incremental adaptations to the landscape, keeping the system in dynamic equilibrium [43]. When the dominant way of doing something is destabilized due to events at the regime or landscape level, there is an opportunity for niche innovations to spread in the regime. Urban transition involves the creation, replacement and standardization of urban regimes and practices [27,43]. Therefore, the smart city involves complex transformational processes stemming from profound modifications around the hard—infrastructure and resources—and soft—human and social capital, knowledge, social innovation, etc.—components of existing urban regimes [1,3,27]. The MLP shows that changing to other forms of mobility is a challenge that not only responds to individual decisions, but also to changing the structures of the socio-technical regime [13,36]. 

**Proposition** **1.***Landscape and regime conditions are combined to explain the readiness of cities in order to implement mobility innovations*.

Each city drives its transformation processes and development levels in different directions according to its contextual conditions, needs and resources [50]. In fact, each city designs its own smart city strategy depending on its available financial resources, cultural background and intrinsic level of development [56]. Among the factors explaining disparities in mobility choices is the income level of citizens [57]. Being wealthier gives local governments an advantage because initiatives and projects that drive a city’s competitiveness, in terms of how smart and sustainable it is, are usually capital intensive [58]. Gross domestic product (GDP) per capita is an expression of wealth, an important predictor of transitions towards sustainability [56]. Higher GDP allows for greater financial resources to be raised for investment, and cities with higher GDP growth rates can attract those who want to improve their living standards and could help to increase the smartness of an urban environment [56]. GDP per capita is an expression of wealth positively associated with a city’s smartness [56,58]. 

**Proposition** **2.***The combination of landscape and regime conditions that explain the readiness of cities to implement mobility innovations differs between the contexts in which the city is located as measured by GPD per capita*.

### 2.3. Data

The development of this work was based on the indicators presented in the Urban Mobility Readiness Index 2021 for a sample of 60 cities from different regions. Geographically diverse cites, ranging from sprawling megacities to more compact cities or to fast-developing metropolises, were examined. They present distinct mobility challenges and display varied solutions that they are pursuing. To rank the cities, the Index employs five dimensions that are the result of 57 key performance indicators that identify cities that excel in meeting their future mobility needs [59]. The Oliver Wyman indices have been used as benchmarks in several academic studies [13,60,61,62,63]. The components used at the different levels are presented in Table 2. 

### 2.4. Method

Urban sustainability is a complex system affected by economic, social and environmental dimensions, with interactions between them [64,65]. Therefore, by focusing the analysis on the conjunctural causation—also referred to as causal complexity—that determines the presence of a given solution, QCA was chosen. This causal complexity offers a realistic description of social phenomena [66]. QCA is an appropriate technique for applying MLP to sustainable transitions [13,35]. Similarly relevant are the epistemological assumptions that allow it to explain in depth the phenomena under study, such as asymmetry, equifinality and the aforementioned conjunctural causation. For all these reasons, research linked to QCA is emerging in the academic sphere [67]. To this is added the fact that it is ideal for works with a small or medium sample size—close to 50 cases [68,69]. Applying QCA, it is possible to systematically analyze a set of cases in order to determine causal patterns in the form of necessary and sufficient conditions between a set of conditions and a certain result [69].

QCA is a technique that integrates elements of qualitative and quantitative analysis. It uses Boolean logic and set analysis to identify combinations of conditions that explain the presence of a result. Initially, analysis was focused on crisp variables—those that can only take the value of 0 or 1—hence its name csQCA (crisp set QCA). Fuzzy set QCA (fsQCA) is a variant of QCA that allows the calibrated values to oscillate between 0 and 1, so it can identify both differences in kind and differences in degree. fsQCA is an interesting complement to regression analysis as it focuses on the analysis of effects from a different perspective. While regression methods find the net effect of the variables, fsQCA offers a set of combinations—called recipes or complex causal patterns—that explain the result [70,71]. The advantage of QCA over regression-based techniques is that they establish relationships between subsets of conditions in order to explain the relationships [69]. In this way, it is possible to identify both the causes that explain the outcome and the impact of each of them [71]. Complementing the assessment of the explanatory and predictive power of the model offered by statistical techniques, QCA generates deeper insights into the relationship between variables, offering a way to reach better management conclusions [72,73]. 

## 3. Results

The first step in the analysis was to calibrate the ratings received by the components of the landscape, regime and niche dimensions using the 95th and 5th percentiles for complete inclusion or exclusion and the mean for the crossover point [13]. To ensure that untenable assumptions were not incorporated, enhanced standard analysis was used [74,75]. Additionally, the intermediate solution was chosen for the presentation of the results. The intermediate solution is considered the preferred solution in QCA as it is the most successful in discovering robust sufficiency [76,77], and allows a more comprehensive and meaningful interpretation [73]. With the intermediate solution, all those remainders that are in line with the directional expectations are incorporated in the minimization process, those theoretical affirmations that guide the investigation by proposing the way in which the causal conditions could affect the presence of the outcome [75,78]. The intermediate solution lies between the conservative and parsimonious solutions because it includes logical remainders, but only those that are considered sensible [75,79]. Several authors have considered the decision to filter out all those remainders that contradict the theory as sensible [78,79].

### 3.1. Analysis of the Interaction between the Conditions of the MLP

In order to explain how the different conditions of the MLP interact in the suitability for implementing mobility innovations, the two-step protocol was applied. This is because, between landscape and regime conditions, a distinction can be made between remote (landscape) and proximate (regime) factors. This protocol establishes a first step corresponding to a necessity analysis for the remote conditions, and a sufficiency analysis for the proximate conditions and those that have been found to be necessary [74]. RStudio’s SetMethods package was used for the analysis.

Within the first step of the two-step protocol, the necessary conditions were analyzed. A condition is necessary for the outcome if whenever the outcome is present, the condition is also present [75]. As a result, there is no atomic necessary condition for the suitability of implementing sustainable mobility solutions (OVE), but one for its negation (~OVE): ~INN (inclN = 0.947; covN = 0.845; RoN = 0.862). In necessity, consistency (inclN) is a measure of the extent to which empirical evidence is in line with a set relation; coverage (covN) expresses the difference in size between the condition and the outcome sets; and relevance of necessity (RoN) is the more conservative measure of the empirical relevance [75]. 

Enhanced standard analysis was applied to establish sufficient conditions. A condition is sufficient if its presence explains the appearance of the result. A consistency of 0.85 and one case per conjunction were used to create the truth table. No contradictory simplifying assumptions were identified, so only implausible counterfactuals had to be eliminated. All atomic regime conditions that were necessary from the necessity analysis were taken as directional expectations. For OVE, these are INF (inclN = 0.911; covN = 0.959; RoN = 0.958), SIM (inclN = 0.917; covN = 0.906; RoN = 0.895), MAT (inclN = 0.928; covN = 0.890; RoN = 0.872) and SEF (inclN = 0.904; covN = 0.951; RoN = 0.950). For ~OVE, they were as follows: ~INF (inclN = 0.953; covN = 0.899; RoN = 0.915) and ~SEF (inclN = 0.944; covN = 0.890; RoN = 0.908). 

The enhanced intermediate solution for OVE results in SIM + INF*MAT*SEF -> OVE. Thus, the solution is composed of two terms, in which the different regime-level conditions considered in the model are presented. Both terms have high parameters, SIM (inclS = 0.906, PRI = 0.872, covS = 0.917, covU = 0.140) and INF*MAT*SEF (inclS = 0.997, PRI = 0.995, covS = 0.819, covU = 0.140). Consistency—also called inclusion score—is the ratio of cases that, presenting the conditions of the solution, explain the result under analysis. In fsQCA, the consistency of the solution shows a similar role as the test ratios. Coverage is the ratio of cases that, presenting the results, are explained by the solution. The coverage of a solution measures the ability of the recipe to explain all of the observations and can be assimilated to R^2^ [70]. Coverage is the parameter that explains the empirical relevance of the solution. The coverage of the solution is fewer than the sum of the coverages of the different conjunctions that make up the solution, due to the existence of overlaps between said conjunctions. For this reason, unique coverage, the degree of coverage attributable to a single condition, is also presented [75]. For example, in the case of OVE, the coverage of the solution is 0.959, less than the sum of the coverages of the two conjunctions that compose it: SIM = 0.917; INF*MAT*SEF = 0.819. This is due, as indicated above, to the existence of overlapping cities. The cities explained by each solution are presented in the footnote. For example, in the case of OVE Oslo it is explained by both conjunctions. As can be seen (Table 3), both solutions have low unique coverage, demonstrating the existence of numerous overlaps between them.

In the case of ~OVE, the enhanced intermediate solution is ~INN*~INF*MAT + ~INN*~INF*~SEF + ~INN*~SIM*MAT + ~INN*MAT*~SEF + ~INF*~SIM*MAT*~SEF -> ~OVE (Table 4). The solution consists of five conjunctions with acceptable parameters: ~INN*~INF*MAT (inclS = 0.909, PRI = 0.688, covS = 0.381, covU = 0.001), ~INN*~INF*~SEF (inclS = 0.976, PRI = 0.964, covS = 0.873, covU = 0.496), ~INN*~SIM*MAT (inclS = 0.944, PRI = 0.806, covS = 0.342, covU = 0.007), ~INN*MAT*~SEF (inclS = 0.914, PRI = 0.702, covS = 0.390, covU = 0.005), ~INF*~SIM*MAT*~SEF (inclS = 0.961, PRI = 0.867, covS = 0.343, covU = 0.018). It is also worth noting how the conjunction ~INN*~INF*~SEF explains a good proportion of the cases, as it has a covU of 0.496. Beyond the presence of the negation of conditions at the regime level, it is worth noting the presence of MAT among the conditions that explain ~OVE. Likewise, the necessary condition ~INN appears in many of the solutions.

The graphical representation (Figure 1) of the solutions shows the existence of no deviant coverage cases for the solution of OVE and ~OVE. However, some deviant cases appear consistent in kind. 

To establish the robustness of the results and increase the credibility of the conclusions, existing tests were performed [34,75]. The calibration range test shows how modifying the different calibration benchmarks would have no impact on the results within the ranges shown in the table below (Table 5). There would have been an impact on the results if the number of cases required was modified above 1; however, it would not have made sense to set an “n cut” higher than 1 because of the sample size. Likewise, changing the required consistency level beyond 0.85 would also have an impact on the results.

In the table above (Table 5), we observe the fit-oriented robustness, comparing fit parameters for the initial solution (IS), the robust core (RC) and the minimum and maximum test set. Since all parameters score above 0.7, except for RF_SC:minTS, which is 0.695 for ~OVE, we consider the results robust. In the case-oriented robustness, the robustness case ratio for deviant cases consistency presents low levels, indicating a reduced number of robust deviant consistency cases compared to all deviant consistency cases in kind (robust, shaky and possible). Additionally, the rank indicates the existence of shaky and deviant cases.

### 3.2. Analysis Cluster

To establish whether the solutions obtained are generalizable across cities with a different GPD per capita, cluster analysis was performed to indicate whether the solution obtained for the pooled data using QCA was appropriate for each sub-population’s data. 

Unlike other techniques, in SetMethods, cluster is a diagnostic tool for clustered data. In other words, it is not intended to establish data grouping, but rather it is based on previous data grouping. In our case, we start from the allocation that is made of the cities under analysis in the index used, which groups cities based on their GDP in the following groups: high, upper-mid, lower-mid, and low [59]. In this case, the grouping used for the data would be correct; otherwise, it would not be correct [74]; that is, it is not about identifying a new solution for each of the clusters, but rather about establishing the capacity of the solution to explain the behavior of the different components of said cluster. With cluster analysis “we can set the diagnostic of how the solution holds throughout the different units and clusters by just imputing the solution” [75] (p. 527).

In the table above (Table 6), we observe the cluster analysis results for the explanation of OVE. The rows “between …” show consistency and coverage values of the terms for each cluster. These are the values that we would have obtained if the analysis had been performed for each cluster separately [75]. The pooled consistency indicates the overall consistency observed in the data when clusters are not taken into account. The between consistency is a measure of the cross-sectional consistency for each cluster [80]. From this finding, we can see how either of the two solutions would explain OVE on the basis of consistency, coverage and, in particular, the value taken by the distance. The between distance is defined as the Euclidean distance between the normalized dimensional vector of the consistencies [80] The results for the cluster analysis of ~OVE are also presented. For the case of the clusters corresponding to low, lower-mid and upper-mid GPD per capita, the conjunction ~INN*~INF*~SEF is the one that explains the ~OVE. This conjunction also explains the case of cities characterized by high GPD per capita, as it is the only one that simultaneously fulfils both consistency and coverage. While the conjunction ~INN*~SIM*MAT has high consistency, it does not meet the coverage criterion. The reverse is true for the ~INN*MAT*~SEF conjunction.

### 3.3. Regression Analysis

To establish the individual net effects, a multiple linear regression was performed, in which OVE was explained on the basis of INF, SIM, MAT, SEF and INNThe significance of the regression coefficients evaluated with Student’s *t*-test and the overall goodness of the regression model with R^2^ [70]. This regression has an adjusted coefficient of determination of 0.999. That value shows the explained percentage of the variance of the regression in relation to the variation of the explanatory variable, taking into account the sample size and the inclusion of variables. In the case of the present model, it is high, being very close to the maximum value that it can reach. The F-value of the model is 2,521,274.088 with five degrees of freedom. The F is linked to an ANOVA analysis in which the variation in the regression is analyzed; that is, how far the data are from the estimate. The resulting equation shows that the corresponding probabilities are the resulting OVE = 0.031 (0.221) + 0.251INF (0.000) + 0.249SIM (0.000) + 0.200MAT (0.000) + 0.200SEF (0.000) + 0.100INN (0.000). The value shown by the t-statistic for the five conditions presented shows that there is statistical relevance. In this way, we can affirm that the relationship between the dependent and independent variable is not due to random probability. Thus, we can see that all five conditions included in the regression are significant, with INF and SIM having the largest impact. In no cases did the standardized residuals present an absolute value greater than 3, which would mean that it could negatively influence the analysis. Of the 60 standardized residuals, the highest absolute value is 2246 and only in three of the cases do they reach 2 in absolute value. To analyze multicollinearity, the Variance Inflation Factor was used. In three of the cases, a moderate value was obtained (SIM = 3.851, MAT = 4.291, INN = 4.190), which implies that the situation is not serious enough to require attention. In the other two cases, the result is severe, so the estimation of the coefficients and the sp values could not be reliable (INF = 8.621, SEF = 7.558). However, we must remember that using indices for a multiple regression can generate situations of high multicollinearity. Likewise, the coefficients have been taken only as a reference to assess their interpretation regarding conjunctural causation.

## 4. Discussion

Urban mobility has great implications for the health and sustainability of cities and the population that resides in them. This situation deserves our attention for two main reasons [2,7,54]: the growing concentration of the population in urban areas and the COVID-19 pandemic. The disruption caused by the pandemic must become an opportunity to make changes towards achieving sustainability. These crises affect the urban environment and require transformations of urban structures, not only in terms of physical spaces but also in terms of operations and behavior [54]. The emerging strategies in urban mobility are transformative in line with the principles of smart growth and sustainable development [54]. For this reason, it is of interest to analyze the ways in which cities are ready for the implementation of innovations in mobility—innovations that, if they occur, would put cities on the path toward achieving SDG3, SDG11, and SDG12.

We begin by analyzing the role of INN as a landscape condition. The literature has argued that increasing innovation can help cities to prepare for the transformation into a smart city by fostering cutting-edge technologies and ideas, stimulating business growth and generating employment opportunities [27,49]. However, in our case, it has only been ~INN’s role as a necessary condition for ~OVE, with no such role for OVE. We must remember that INN is linked to the way in which a city takes advantage of local talent and resources to drive technological advances. This definition is in line with those who indicate that the smart city must suppose an incorporation of the human capital variable with the relevance that has already been given to technological elements in the definition of the smart city [5,6,21,22]. Likewise, the advances in INN must be reflected in the achievement of SDG11. Based on the results achieved, we see how opting for a current definition of a smart city—combining human and technological components—is not necessary to prepare for the adoption of innovations in mobility (OVE), although its absence is necessary for its negation (~OVE). In this way, the readiness for the implementation of innovations in mobility is produced as a result of the operation of the conditions at the regime level. In line with what has been stated in previous studies [13,36], the change towards new mobility systems will be determined by changes in the socio-technical regime. Likewise, this approach can confront those who present the regime as a stabilization of trajectories causing maintenance of the status quo [33,42,43].

Since cities are systems with a constant evolution which must be guided by the SDG [23,42,55], it is necessary to try to generate tension between the landscape and the regime so that innovations at the niche level emerge [47]. Therefore, if the INN does not play this role, progress must be made in activating other elements, such as actions aimed at SDG9 linked to industry, innovation and infrastructure. Let us remember that in our model, we considered infrastructure linked to mobility as a regime condition. There remains a wide range of infrastructure and industry in which cities could try to advance.

In the explanation of OVE, in line with [31], the relevant role played by SIM in the transformation of cities is verified, so that it explains the readiness of the city for implementing mobility innovations. This means that mobility not only impacts the achievement of a sustainable urban environment [12,15], but also maximizes the social benefits that these initiatives can bring about, which is related to the city’s readiness for the implementation of mobility innovations (OVE). Since the definition of SIM reflects the generation of employment opportunities, it is a condition that can help achieve SDG8. The authorities must bear in mind the multiplier effect of certain actions. Thus, the commitment to promote jobs linked to the field of mobility will benefit the conditions that determine the readiness of the city to implement innovations in mobility, as well as the benefits that are linked to it.

In the case of OVE, the second solution is that in which INF, MAT and SEF are presented together. The transition processes towards sustainability can be shown through different paths that overlap each other [37]. This reaffirms how urban development is a product of the confluence of infrastructure, and alternative solutions with the potential to disrupt the regime [44]. The SEF component is linked to the coordination that is presented in the system. For this reason, it places us in line with those who affirm that mobility solutions include advanced forms of cooperation with the environment and companies—regime—or that large-scale changes require processes involving a large number of factors that interact to disrupt the status quo [36]. This shared governance places us, once again, in line with SDG11. The link of INF and SEF can present a situation close to those who are committed to a flexible and efficient smart city [21], and those advocating for infrastructures that improve health and reduce emissions [50]. Additionally, while INF is required for the adoption of mobility innovations, it does not seem to play the key role that has been outlined previously. Cities are transforming their infrastructure towards achieving a smarter and more efficient approach to sustainable development [1]. The pressure from the dominant automobile system on infrastructure—worsening urban congestion, increasing travel times and accidents—leads to pressure for the renewal of mobility systems, although these do not rest exclusively on the said infrastructure [20,36]. However, we must remember the positive effects of offering infrastructure and transport systems. These are among the key actions of cities positively impacting the health of their citizens [9]. Likewise, mobility infrastructures are also linked to the fight against inequalities in access to health (SDG3), work (SDG8), and infrastructure (SDG9) [81]. Being able to overcome the blockage that the dominant regime of the automotive sector presents will help initiate changes that will help to achieve SDG7 and SDG12. In relation to SDG12, we recall that achieving the leap to sustainable mobility solutions not only implies offering such modes of transport. To achieve change, and for the niche linked to sustainable mobility to emerge, it is necessary to have a set of agents and technologies that make up the regime.

If we explain ~OVE, ~INN is a necessary condition. Although from the results obtained for OVE, we cannot affirm that INN is among the factors that determine readiness for OVE [27,49], we can indicate that for ~OVE to be present, ~INN must be present. There are multiple solutions and conjunctions that explain ~OVE. In some of them, the union of ~INN and ~INF appears, which can be aligned with those who linked innovation and the use of infrastructures in the modernization processes of cities [4]. Furthermore, the fact that ~INF partly explains ~OVE supports those who point to the centrality of infrastructure in determining the readiness of smart cities.

It seems striking that MAT, depending on the conditions with which it is combined, can determine both OVE and ~OVE. Thus, it appears that MAT is linked, to a greater extent, with the ability to attract a creative population or one with some economic strengths [51], rather than with the decision to choose, or not, to implement mobility-related innovations. The MAT component is linked to smart mobility activation and the availability of public funding. The mere development of protected niches and incentives that support sustainable mobility innovations will not be enough for a profound transformation in urban mobility systems. Existing dominant regimes need to be pressured [20,36]. The transition towards sustainable mobility requires changes in the regulatory framework. Without such a framework, sustainable mobility services will remain a niche that increases—rather than reduces—the number of vehicles in use [36]. In this way, the implications of promoting the niche linked to the implementation of innovations in mobility must be taken into account due to its link with SDG3, SDG7 and SDG 12.

Likewise, although it has been proposed in previous studies that INN and INF are the determinants of urban innovation, in our case, this is not verified. What is found is that ~INN and ~INF occur simultaneously to explain ~OVE. In the present work, based on previous works [1,3,27], we can establish that the negation of readiness to implement innovative solutions in mobility emerges from deep modifications in the hard and soft components.

Cluster analysis shows us the possibility of explaining OVE and ~OVE independently of the context in which it takes place. However, this does not mean that there is only one way for cities to be ready to implement mobility innovations, as there are two options for explaining OVE. A similar situation arises for the explanation of ~OVE, where the values achieved show the suitability of analyzing all cities under analysis as a whole. The fact that the most enriched urban environments are not more proactive in investing in smart city projects can be explained by the fact that, as a result of this level, they are no longer interested in developing new projects [56].

Finally, when analyzing the isolated effect of the conditions, the impact is similar, and there are notable differences when analyzing the joint effects of the conditions. In fact, INN, which obtains the lowest weight when establishing the isolated effects, comes to act as a necessary condition. Thus, the suitability of combining QCA and regression analysis is confirmed [71,72]. Additionally, the factors that explain OVE are not the inverse of those that explain ~OVE. In this way, the authorities should bear in mind the disparate effect that opting for the combination of different conditions would entail. When we compare the results obtained from the multiple regression with those corresponding to QCA, elements of notable relevance emerge. The regression shows us how OVE is significantly determined by the five components that act as independent variables, with a greater weight for INF and SIM. However, we draw two great lessons from the application of QCA. In the first place, there are two paths through which the OVE can be achieved, implying different combinations of conditions. In fact, in the first explanatory solution of OVE, only one condition appears, which, moreover, does not appear in the second solution. This element links to the second lesson, which states that the effect of the conditions depends on those with which they are combined. In this way, as we have already indicated, MAT could explain OVE or ~OVE depending on those conditions with which it occurs simultaneously.

## 5. Conclusions

This paper has been developed with the aim of identifying how the conditions that determine the level of readiness among cities for adopting urban mobility innovations could be combined, responding to the request of the literature [2,28]. This objective is relevant due to the impact that the population concentration process in urban areas and the mobility solutions linked to it have on the health of citizens; it has even been claimed that cities are fundamental to the achievement of the SDGs. More specifically, this study aimed to address two research gaps.

The first gap was identifying the combinations of factors that determine the level of readiness of cities to implement sustainable mobility innovations. The results show such combinations of factors for both OVE and ~OVE. Thanks to conjunctural causation, we have seen how, except in the case of the SIM effect for OVE generation, the conditions combine at least in groups of three to explain the phenomena under study. Thus, we can state, in line with [30], how different types of factors combine in the preparation of smart cities.

As a second research gap, we have analyzed in different contexts the conditions that determine the implementation of mobility innovations in the smart city domain. Firstly, because all cities can be grouped together in the explanation of OVE and ~OVE, it does not seem that the context plays a determining role in the way in which the conditions determining their emergence combine. However, this does not mean that there is a single formula common to all cities. As we have already indicated, there are various combinations of factors that explain the phenomena under study in this paper.

Just as mobility has not been established in the same way all over the world [26], the readiness to implement innovations does not necessarily have to be established in a strict way either. Thus, thanks to the application of QCA, we have been able to analyze causal relationships between conditions acting at different levels [2]. More precisely, we can observe the benefits derived from its epistemological assumptions. We observe the asymmetry, whereby OVE and ~OVE are explained by different combinations of conditions. Additionally, thanks to equifinality, the different ways in which the same phenomenon can be explained are exposed. Finally, thanks to joint causation, we observe how certain conditions (MAT) can lead to OVE and ~OVE depending on those with which they are combined. Linked to joint causation, we find how the building blocks of STTs in urban mobility rest on the convergence of factors [33].

Among the limitations of this work, it is shown how modifying the inclusion in the truth table would have altered the results obtained. Additionally, if we look at the case-oriented robustness parameters, the presence of shaky and possible cases is shown.

## Figures and Tables

**Figure 1 ijerph-19-07197-f001:**
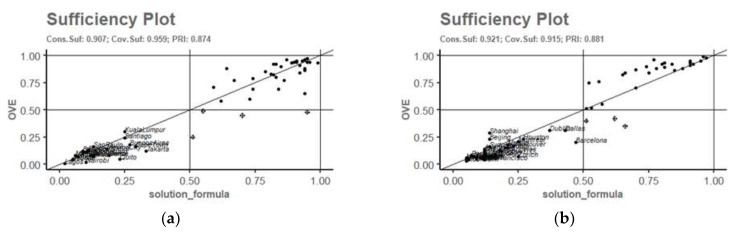
Graphical representation of solutions (**a**) Pimplot OVE; (**b**) Pimplot ~OVE.

**Table 1 ijerph-19-07197-t001:** SDGs linked to sustainable mobility.

SDG	Description
3	Good health and well-being
7	Affordable and clean energy
8	Decent work and economic growth
9	Industry, innovation and infrastructure
11	Sustainable cities and communities
12	Responsible consumption and production

**Table 2 ijerph-19-07197-t002:** Level and conditions applied.

Level	Condition	Description
Landscape	Innovation (INN)	How well does the city leverage local talent and resources to drive technological advances?
Regimen	Infrastructure (INF)	Has the city developed robust infrastructure and expanded connectivity to support future mobility?
	Market Attractiveness (MAT)	How well does the city engage the private sector and secure diverse investments to build out mobility?
	System Efficiency (SEF)	How well does the municipal government coordinate and enhance the city’s mobility network through things like traffic management systems?
	Social Impact (SIM)	Does the city maximize societal benefits like mobility-related employment or airport arrivals while minimizing harmful qualities like poor air quality?
Niche	Readiness (OVE)	Readiness as an indication of its future mobility capacity.

**Table 3 ijerph-19-07197-t003:** Enhanced intermediate solution OVE.

	inclS	PRI	covS	covU	Cities Covered
SIM	0.906	0.872	0.917	0.140	Doha, Abu Dhabi, Dubai, Milan, Moscow, Zurich; Istanbul, Berlin, Atlanta, Dallas, Houston, SanFrancisco, Chicago, NewYork, LosAngeles, Boston, Sydney, Helsinki, Dublin, Toronto, Vancouver, Madrid, Montreal, Munich, Oslo, Amsterdam, Seoul, Stockholm, Washington.D.C., Paris, Barcelona, London, Singapore, Tokyo, HongKong
INF*MAT*SEF	0.997	0.995	0.819	0.041	Warsaw, Beijing, Shanghai, Berlin, Atlanta, Dallas, Houston, SanFrancisco, Chicago, New York, LosAngeles, Boston, Sydney, Helsinki, Dublin, Toronto, Vancouver, Madrid, Montreal, Munich, Oslo, Amsterdam, Seoul, Stockholm, Washington.D.C., Paris, Barcelona, London, Singapore, Tokyo, Hong Kong
Solution	0.907	0.874	0.959		

**Table 4 ijerph-19-07197-t004:** Enhanced intermediate solution for ~OVE.

	inclS	PRI	covS	covU	Cities Covered
~INN*~INF*MAT	0.909	0.688	0.381	0.001	Dubai, Milan, Moscow
~INN*~INF*~SEF	0.976	0.964	0.873	0.496	Johannesburg, Jakarta, Bangkok, Quito, Jeddah, Riyadh, Buenos Aires, Cape Town, Nairobi, Rio de Janeiro, Sao Paulo, Lagos, Manila, Casablanca, Santiago, Mexico City, Cairo, Lima, Delhi, Bogota, Mumbai, Doha, Abu Dhabi, Dubai
~INN*~SIM*MAT	0.944	0.806	0.342	0.007	Warsaw
~INN*MAT*~SEF	0.914	0.702	0.390	0.005	Dubai, Istanbul
~INF*~SIM*MAT*~SEF	0.961	0.867	0.343	0.018	Kuala Lumpur
Solution	0.921	0.881	0.915		

**Table 5 ijerph-19-07197-t005:** Robustness test.

Robustness Calibration Range				
		Lower Bound	Threshold	Upper Bound
INF	Exclusion	NA	34.2	58.2
	Crossover	47.4	58.4	60.4
	Inclusion	59.3	81.3	NA
SIM	Exclusion	18.6	34.6	56.6
	Crossover	46.9	56.9	56.9
	Inclusion	57.7	72.7	NA
MAT	Exclusion	NA	19.9	52.9
	Crossover	52.5	53.5	54.5
	Inclusion	54.4	73.4	NA
SEF	Exclusion	NA	34.4	45.4
	Crossover	49.2	53.2	56.2
	Inclusion	53.6	71.6	NA
INN	Exclusion	-8.4	5.6	38.6
	Crossover	27.6	39.6	41.6
	Inclusion	40.1	75.1	NA
Raw Consistency Test		0.85	0.85	0.85
N.Cut range		1	1	1
**Robustness parameter OVE**				
Fit_Oriented	RF_cov: 0.743 RF_cons: 0.989 RF_SC_minTS: 0.736 RF_SC_maxTS: 0.841			
Case_Oriented	RCR_typ:735 RCR_dev:0.25 Rank:4			
**Robustness parameter ~OVE**				
Fit_Oriented	RF_cov: 0.714 RF_cons: 0.975 RF_SC_minTS: 0.695 RF_SC_maxTS: 0.772			
Case_Oriented	RCR_typ:0.654 RCR_dev:0.071 Rank:4			

**Table 6 ijerph-19-07197-t006:** Cluster analysis.

	Result: OVE		Result: ~OVE				
	SIM	INF*MAT*SEF	~INN*~INF*MAT	~INN*~INF*~SEF	~INN*~SIM*MAT	~INN *MAT*~SEF	~INF*~SIM *MAT*~SEF
**Consistencies**							
Pooled	0.906	0.997	0.909	0.976	0.944	0.914	0.961
Between High	0.936	1.000	0.774	0.823	0.926	0.799	0.884
Between Low	0.807	1.000	1.000	1.000	1.000	0.994	1.000
Between Lower-mid	0.943	1.000	0.909	0.997	0.979	0.919	0.975
Between Upper-mid	0.881	0.987	0.951	0.985	0.875	0.935	0.962
**Distances**							
From Between to Pooled	0.031	0.003	0.046	0.039	0.026	0.039	0.023
**Coverages**							
Pooled	0.917	0.819	0.381	0.873	0.342	0.390	0.343
Between High	0.936	0.826	0.885	0.885	0.595	0.897	0.667
Between Low	0.849	0.751	0.276	0.915	0.256	0.282	0.253
Between Lower-mid	0.903	0.826	0.330	0.851	0.322	0.327	0.326
Between Upper-mid	0.935	0.833	0.420	0.824	0.412	0.444	0.387

## Data Availability

Not applicable.

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
