# Peer review of "The Transition of Cities towards Innovations in Mobility: Searching for a Global Perspective"

_ijerph, 2022, doi:10.3390/ijerph19127197_

Round 1

Reviewer 1 Report

The article has a very interesting title and promises a lot, although the abstract does not. The abstract of the article is not very communicative and the abstract does not show what the research authors conducted and for what purpose.

The article describes interesting research and approach in a very non-communicative way. It's hard to read  because of both, the language and the way of narration. There is a lot of repetitions and little details in the context.

The Introduction part of the article is a bit wordy, the authors wrote a lot, but the content repeats itself, and the following sentences do not add anything new. This makes the article difficult to read and it is difficult to know exactly what the authors actually did in their study. The use of kay phrases, such as research questions, or the purpose of the article would help much to organize the content.

The authors devote a lot of attention to Smart City without defining it, and the title indicates that mobility and innovation in mobility were the subject of the research. This part is not explained at all in the introduction. Due to the quite diverse approach to the concept of smart city in literature, the authors should be very precise in defining the terms they use. On the other hand, the entire Introduction section  and "The urban transition towards mobility innovations" section contain many general, not specified statements, i.e. innovation, smart, sustainable.

First research gap is about to implement mobility innovations (lines 69-71) wile the second sustainable research gab is about adoption of the smart city approach (lines 90-92). The authors do not define what they mean by innovative mobility in cities. I have a question for authors: for example, one of the new concepts for the cites is walkable cities or 15-minutes cicites based on walking - is walking the innovative mobility solution for you?

The valuable content of the article starts only from line 162, but should be placed in section 3 Materials and Methods, as it partially includes the method description. Although I do not understand subsection 3.1. Model. Proposals 1 and 2 sound like theses, do I understand correctly that these are proposals for a method to measure cities' readiness for innovation in mobility? In section 3, I also miss the link between the QCA method and the MLP approach. Maybe in section 3 it would be nice to describe the research procedure - this procedure is described a bit in section 4 Results.

In Discussion section there is a statement: “As previously stated, urban mobility has great implications for the health and sustainability of cities…” (line 452) - no there is no such clear statement in previous sections of the article. The Discussion section is only one which explains the research author did. I would expect that in Methods and results section and in Discussion more a reference to the results of similar studies by other authors or to previous studies by these authors.

First sentence in Conclusion (line 545) is aim of the article - finally! Please, communicate the aim of the study in Abstract, Introduction and Material and Methods sections.

Author Response

Reviewer 1

First of all, we would like to acknowledge the comments and indications made by the reviewer. As you can see, in the text of the article we have tried to respond to your suggestions, as well as those of the rest of the reviewers. We will now explain the question to each of its indications.

The article has a very interesting title and promises a lot, although the abstract does not. The abstract of the article is not very communicative and the abstract does not show what the research authors conducted and for what purpose. Following the editor's recommendations, the title of the article has been changed. Likewise, the wording of the abstract has been modified, incorporating both what was done and the purpose of the work.

The article describes interesting research and approach in a very non-communicative way. It's hard to read  because of both, the language and the way of narration. There is a lot of repetitions and little details in the context. As you can see in the text of the work, there have been numerous elements that have been extracted from the body of the article. Attempts have been made to identify the aforementioned duplications. Likewise, a proof-reading of the article has been carried out.

The Introduction part of the article is a bit wordy, the authors wrote a lot, but the content repeats itself, and the following sentences do not add anything new. This makes the article difficult to read and it is difficult to know exactly what the authors actually did in their study. The use of kay phrases, such as research questions, or the purpose of the article would help much to organize the content. As stated above, elements of the abstract have been removed in order to synthesize the contents. Likewise, the use of kay phrases has been reinforced, as requested. More specifically, the research question, purpose and gap of the work appear.

The authors devote a lot of attention to Smart City without defining it, and the title indicates that mobility and innovation in mobility were the subject of the research. This part is not explained at all in the introduction. Due to the quite diverse approach to the concept of smart city in literature, the authors should be very precise in defining the terms they use. On the other hand, the entire Introduction section  and "The urban transition towards mobility innovations" section contain many general, not specified statements, i.e. innovation, smart, sustainable. As is well indicated, many elements were not clearly delimited in the Introduction. Elements that, on occasions, were presented in the section "The urban transition towards mobility innovations". In line with what was suggested in a later comment, part of the content of the mentioned section has been distributed between the Introduction (the elements focused on the delimitations) and the “Matherials and Methods” section. Regarding the definition of the smart city, the introduction indicates that it is committed to the smart city approach supported by the SDGs.

First research gap is about to implement mobility innovations (lines 69-71) wile the second sustainable research gab is about adoption of the smart city approach (lines 90-92). The authors do not define what they mean by innovative mobility in cities. I have a question for authors: for example, one of the new concepts for the cites is walkable cities or 15-minutes cicites based on walking - is walking the innovative mobility solution for you? The presentation of the research gap has been modified to achieve greater consistency between them. Regarding the question posed by the reviewer, I consider that the innovative mobility solution depends (as indicated in the article) on the context. In a large city, possibly the solution can be linked to a greater extent by a combination of sustainable solutions (in line with MaaS). In a city with a smaller area, possibly part of the solution can be linked to the "walkable city".

The valuable content of the article starts only from line 162, but should be placed in section 3 Materials and Methods, as it partially includes the method description. Although I do not understand subsection 3.1. Model. Proposals 1 and 2 sound like theses, do I understand correctly that these are proposals for a method to measure cities' readiness for innovation in mobility? In section 3, I also miss the link between the QCA method and the MLP approach. Maybe in section 3 it would be nice to describe the research procedure - this procedure is described a bit in section 4 Results. Following your suggestion, the article has been restructured. Exposing the model and the method, distributing part of the contents of section 2 between the introduction and the new section 2. Likewise, the relationship between QCA and MLP has been exposed, since it has been previously applied. The main reason for its use is that QCA allows measuring conditions at different levels, focusing on the interactions between them. Likewise, propositions are presented since in QCA it is recommended not to use hypotheses but rather propositions. We hope that the profound reformulation of section 2 meets the expectations of the reviewer.

In Discussion section there is a statement: “As previously stated, urban mobility has great implications for the health and sustainability of cities…” (line 452) - no there is no such clear statement in previous sections of the article. The Discussion section is only one which explains the research author did. I would expect that in Methods and results section and in Discussion more a reference to the results of similar studies by other authors or to previous studies by these authors. Agradecemos el comentario y nos ha servido para replantear determinados componentes del trabajo. No obstante, lo habíamos indicado previamente en el artículo cuando, por ejemplo, indicamos “Sustainable mobility plays a key role inarchieving a sustainable urban environment, generating benefits for the well-being and public health of cities [40, 45]”. Esperamos que ahora se exponga claramente lo que los autores hemos hecho. En la discusión, consideramos que se presentan numerosas referencias a otros autores. Esperemos que la nueva versión del trabajo satisfaga las expectativas del revisor.

First sentence in Conclusion (line 545) is aim of the article - finally! Please, communicate the aim of the study in Abstract, Introduction and Material and Methods sections. We appreciate the comment. The purpose has been incorporated in the Abstract, Introcution and Material and Methods sections.

Reviewer 2 Report

Your paper with its goal to find a global perspective for the transitions of cities to smartness is very ambitious. And as the goal is ambitious, it is not easy to reach. You use many references, sometimes you present them in such a shortcut that it changes the meaning, you reference even small details but sometimes forget to clearly state the main issues, etc.  The text is thus very dense to read, and the results and how you have reached are a bit lost and the way to them is not totally clear.

Less is sometimes more, I strongly recommend you to modify the text to present the background and mainly your results in a bit more straightaway form. Particular issues I find the most problematic are below:

Line 110 – to use as a reference for United nations sustainable development goals an article that just mentions some of the goals is strange and not good. If you from the beginning speak about the SDGs of the UN, you should say it at the beginning – line 41. It would be much better to put there a table with the overview of the SDGs, as you use them through the whole text.

Line 171 the phrase that large social-technical systems require government intervention to correct environmental externalities seems to be taken out of context. The original recourse does not mention such a strong statement. You should support it better, or not the present this as a fact.

Line 183-186 is too simplified. Prolonged stages of the landscape level do not help to destabilize the regime. Source 49 talks about landscape pressures, such as climate change, and source 50 says that: “An analysis of the landscape level is not included in this study“- Your sentence is misguiding.

Chapter 3.1

The description of the three levels is not clear and is not in accordance with the sources – e.g. [50]. There in the landscape the land use, utility infrastructure is mentioned, here you have the infrastructures and disposition of urban spaces among the regime conditions, which is strange. The source name as the regime conditions “practices and materials that make up our everyday world, including dominant beliefs, values and norms”

Line 268 – you say that smart cities implementation is consistent with economic development, but not with ecological development [29]. But this paper does not study ecological development, and is only about situation in China. This cannot be so easily generalized. Such a statement is too strong and I do not believe true.

Line 305 – typing error – GPD instead of GDP

Chapter 3.2. As you work with the Oliver Wyman indices, it would be nice to explain their meaning more in depth, not just name them, but just briefly where the values come from.

Line 329 – you should explain the fsQCA abbreviation, not just that it is in the name of some of the resources

Tables 2 and 3 – it is not clear how you created the clusters of the cities having the same intermediate solutions. One can of course guess, but it should be clearly written.

And it is not clear why the intermediate solutions are, as they are. Did you test also other solutions? Why just these solutions for these groups of cities? Etc.

Line 686 – two typing mistakes

Author Response

Reviewer 2

First of all, we would like to acknowledge the comments and indications made by the reviewer. As you can see, in the text of the article we have tried to respond to your suggestions, as well as those of the rest of the reviewers. We will now explain the question to each of its indications.

Your paper with its goal to find a global perspective for the transitions of cities to smartness is very ambitious. And as the goal is ambitious, it is not easy to reach. You use many references, sometimes you present them in such a shortcut that it changes the meaning, you reference even small details but sometimes forget to clearly state the main issues, etc.  The text is thus very dense to read, and the results and how you have reached are a bit lost and the way to them is not totally clear. We appreciate the comment to the reviewer and have tried to lighten the text. The text has been reduced, repetitions have been eliminated and a proof reading has been carried out.

Less is sometimes more, I strongly recommend you to modify the text to present the background and mainly your results in a bit more straightaway form. Particular issues I find the most problematic are below:

Line 110 – to use as a reference for United nations sustainable development goals an article that just mentions some of the goals is strange and not good. If you from the beginning speak about the SDGs of the UN, you should say it at the beginning – line 41. It would be much better to put there a table with the overview of the SDGs, as you use them through the whole text. The commented reference has been removed. Likewise, the table suggested by the reviewer has been included. Finally, the presence of the SDGs in the discussion of the results has been reinforced.

Line 171 the phrase that large social-technical systems require government intervention to correct environmental externalities seems to be taken out of context. The original recourse does not mention such a strong statement. You should support it better, or not the present this as a fact. We have proceeded to remove this phrase.

Line 183-186 is too simplified. Prolonged stages of the landscape level do not help to destabilize the regime. Source 49 talks about landscape pressures, such as climate change, and source 50 says that: “An analysis of the landscape level is not included in this study“- Your sentence is misguiding.  I believe that we have not been able to transfer some of the main messages of the article, so we have proceeded to modify numerous elements of the wording. Respecting the opinion of the reviewer, citation [50] states in his work “It is possible to make sense of transitions towards more sustainable systems with the help of the three levels of stabilization in the structures of the MLP”.

Chapter 3.1

The description of the three levels is not clear and is not in accordance with the sources – e.g. [50]. There in the landscape the land use, utility infrastructure is mentioned, here you have the infrastructures and disposition of urban spaces among the regime conditions, which is strange. The source name as the regime conditions “practices and materials that make up our everyday world, including dominant beliefs, values and norms”. The structure of the work has been modified, to try to transfer the ideas in a clearer way. To try to make it more precise, two additional references have been used. In our case, we have used the infrastructures within the regimen level since they are infrastructures directly linked to mobility. In fact, the discussion specifies the possibility of keeping in mind the general infrastructures (those not linked to mobility) at the landscape level.

Line 268 – you say that smart cities implementation is consistent with economic development, but not with ecological development [29]. But this paper does not study ecological development, and is only about situation in China. This cannot be so easily generalized. Such a statement is too strong and I do not believe true. We appreciate the indication, we have proceeded to eliminate the phrase.

Line 305 – typing error – GPD instead of GDP. Thank you very much for the indication, we apologize for sending the article with typing errors.

Chapter 3.2. As you work with the Oliver Wyman indices, it would be nice to explain their meaning more in depth, not just name them, but just briefly where the values come from. We have incorporated an explanation of the composition of the Index taken as a reference.

Line 329 – you should explain the fsQCA abbreviation, not just that it is in the name of some of the resources. An explanation about fsQCA has been incorporated.

Tables 2 and 3 – it is not clear how you created the clusters of the cities having the same intermediate solutions. One can of course guess, but it should be clearly written. An explanation has been included in the article regarding the way in which the cluster analysis tries to establish the validity of the solution for the different groups of elements that compose them.

And it is not clear why the intermediate solutions are, as they are. Did you test also other solutions? Why just these solutions for these groups of cities? Etc. The choice of the intermediate solution has been explained, supported by citations.

Line 686 – two typing mistakes. Thank you very much for the indication. We apologize for submitting the work with typing mistakes.

Reviewer 3 Report

Dear Authors,

Detecting a suitable plan for urban mobility is one of the main challenges to addressing transition of cities towards innovations in mobility.  In this sense, this paper presents an interesting study in this area.

I globally have some comments to give a general overview according to the nine following points:

i)  The methodology used seems appropriate in the paper. However, in my opinion, The paper structure should be more updated and improved especially the introduction, literature review, and discussion as well as explain more the aim paper.

ii) The main factors of the paper is the landscape, regime, and niche. It is not clear how to define them and what relationships between them there are in terms of structure, and management?

iii) In addition, the last section needs to discuss more concerning assessed factors. The discussion should provide a critical overview of the findings obtained in articulation with the findings of other studies in this field. What are the new findings? Especially plans recommendation approach how will be benefits in the smart city.

iv) The paper needs additional using updated references.

Hoping they will be helpful for your paper.

Author Response

Revisor 3

First of all, we would like to acknowledge the comments and indications made by the reviewer. As you can see, in the text of the article we have tried to respond to your suggestions, as well as those of the rest of the reviewers. We will now explain the question to each of its indications.

Detecting a suitable plan for urban mobility is one of the main challenges to addressing transition of cities towards innovations in mobility.  In this sense, this paper presents an interesting study in this area. Thank you very much for the comment.

I globally have some comments to give a general overview according to the nine following points:

  1. i) The methodology used seems appropriate in the paper. However, in my opinion, The paper structure should be more updated and improved especially the introduction, literature review, and discussion as well as explain more the aim paper. Following your suggestions and those of another reviewer, we have proceeded to modify the structure of the article. I hope that the profound modification made satisfies the expectations of the reviewer.
  2. ii) The main factors of the paper is the landscape, regime, and niche. It is not clear how to define them and what relationships between them there are in terms of structure, and management?

We hope that the new structure of the work will make it possible to present ideas more clearly. In this way, by modifying the structure of the work, we hope that the exposition of the three levels of the MLP (landscape, regime and niche) will be clearer. The interaction between such elements has been transferred to the justification of the first proposition to achieve greater clarity in the exposition.

iii) In addition, the last section needs to discuss more concerning assessed factors. The discussion should provide a critical overview of the findings obtained in articulation with the findings of other studies in this field. What are the new findings? Especially plans recommendation approach how will be benefits in the smart city.

In the last part of the work, the discussion of the results has been reinforced. Both in the discussion and in the conclusions we have tried to reflect on the implications of the new findings. Likewise, in the discussion additional suggestions have been provided for the managers of smart cities.

  1. iv) The paper needs additional using updated references.

As can be seen, some updated publications have been incorporated. However, following the recommendations of another of the reviewers, a substantial reduction in the number of references has been made. We hope that the new range of references that the work presents will satisfy the reviewer.

Hoping they will be helpful for your paper. Thank you very much, your comments have been very enriching. We hope that the new version of the work will be to the liking of the reviewer.

Reviewer 4 Report

The main questions of the research are well addressed. The research gaps are emphasized, the topic is relevant and interesting. Paper is not original, however interesting. Literature review is well done,The paper is not well writte,, the proof-reading is absolutely necessary.

The text is not clear nor easy to read, conclusions are consistent with evidence and arguments presented in this paper

Author Response

Revisor 4

First of all, we would like to acknowledge the comments and indications made by the reviewer. As you can see, in the text of the article we have tried to respond to your suggestions, as well as those of the rest of the reviewers. We will now explain the question to each of its indications.

The main questions of the research are well addressed. The research gaps are emphasized, the topic is relevant and interesting. Paper is not original, however interesting. Literature review is well done,The paper is not well writte,, the proof-reading is absolutely necessary. Thank you very much for the indication. The work, after the first revision, has been subjected to a proof-reading.

The text is not clear nor easy to read, conclusions are consistent with evidence and arguments presented in this paper. The body of the paper has been substantially modified to make the paper easier to read.

Round 2

Reviewer 1 Report

I am satisfied with authors response for review. The authors have improved article scientific soundness much.

Additional comment to authors - please be more careful with sending the response to the reviewer. One of your response has been left in Spanish. Of course there are google translator and I know a little Spanish, but that is not fair to leave response in Spanish when the article language is English. 

Author Response

Reviewer 1

I am satisfied with authors response for review. The authors have improved article scientific soundness much. We appreciate your comment. Thanks to your contributions, we consider that the work is of higher quality. Likewise, we thank you for the constructive tone in which you have carried them out.

Additional comment to authors - please be more careful with sending the response to the reviewer. One of your response has been left in Spanish. Of course there are google translator and I know a little Spanish, but that is not fair to leave response in Spanish when the article language is English. We apologize for the mistake. Also, we reiterate our thanks for the way in which you have developed the review.

Reviewer 2 Report

I like to see you have improved your paper significantly in the introductory chapters where now the references make sense and you explain all you use further on. Unfortunately this does not apply to the description of your method and results.

In your response you say: “An explanation has been included in the article regarding the way in which the cluster analysis tries to establish the validity of the solution for the different groups of elements that compose them.” And that “The choice of the intermediate solution has been explained, supported by citations.” However in this regard I still see huge space for improvements, your changes in this regard were rather small.

From the description of the table 6 in page 10 it seems, the groups are according the GDP per capita. But (if ever) you are talking about the groups of cities, this cannot be, as some cities appear in more groups. And the meaning of “between high” “between lower-mid” etc. in table 6 is also not clear.

Simply, explain why you cluster the cities in these groups as indicated in page 8 and why you choose the particular intermediate solutions for them as indicated in table 4.

(It looks like table 8 explains it, but should not table 8 then be antecedent to table 4?) Really I do not know, if you have this information in your paper somewhere, definitely it is not clear.

And by the way, you speak about 60 cities, but you do not have 60 of them in page 8.

And please check this sentence in the abstract:  “The explanation of the readiness of cities to implement mobility innovations differs from the explanation of the negation of said readiness.” Seems to miss something – I would expect from.... to...

Author Response

Reviewer 2

I like to see you have improved your paper significantly in the introductory chapters where now the references make sense and you explain all you use further on. Unfortunately this does not apply to the description of your method and results.

We appreciate your comment on the first sentence of the paragraph. We consider that the suggestions made by the reviewers have helped to increase the quality of the work. Regarding the second sentence of the paragraph, we will try to answer you in the precise comments you have made.

In your response you say: “An explanation has been included in the article regarding the way in which the cluster analysis tries to establish the validity of the solution for the different groups of elements that compose them.” And that “The choice of the intermediate solution has been explained, supported by citations.” However in this regard I still see huge space for improvements, your changes in this regard were rather small.

The explanation of the development of the cluster analysis has been deepened, as can be seen in the responses to the following comments.

Regarding the choice of the intermediate solution, more detail has been incorporated about what it consists of.

With the intermediate solution, all those remainders that are in line with the directional expectations are incorporated in the minimization process, those theoretical affirmations that guide the investigation by proposing the way in which the causal conditions could affect the presence of the outcome [75, 78]. The intermediate solution lies between the conservative and parsimonious solutions because it includes logical remainders, but only those that are considered sensible [75, 79]. Several authors have considered sensible the decision of filter out all those remainders that contradict the theory [78, 79]”.

From the description of the table 6 in page 10 it seems, the groups are according the GDP per capita. But (if ever) you are talking about the groups of cities, this cannot be, as some cities appear in more groups. And the meaning of “between high” “between lower-mid” etc. in table 6 is also not clear.

From the question raised by the reviewer, it seems that he refers to the fact that in table 6 cities appear in “more groups”, that is, not in a single group. Each city appears in a single group, based on its GDP per capita. The only case in which the cities are listed is in the footnotes corresponding to each solution. In the response to a following comment, this element has been taken into account, justifying itself in the difference between coverage and single coverage

It has been explained the meaning of “between”: The rows “between ...” show consistency and coverage values of the terms for each cluster. These are the values ​​that we would have obtained if the analysis had been performed for each cluster separately [75].

Simply, explain why you cluster the cities in these groups as indicated in page 8 and why you choose the particular intermediate solutions for them as indicated in table 4.

Proposition 2 states the following: “The combination of landscape and regime conditions that explain the readiness of cities to implement mobility innovations differs between the contexts in which the city is located as measured by GPD per capita”. So, to analyze if cities with different GPD requires different solutions, the cities were grouped based in its GDP in the following groups: high, upper-mid, lower-mid, and low.

The rationale for using the intermediate solution has been stated in the response to a previous comment.

(It looks like table 8 explains it, but should not table 8 then be antecedent to table 4?) Really I do not know, if you have this information in your paper somewhere, definitely it is not clear. Before formulating the answer, I am going to make some clarifications to be able to do it. First of all, there is no table 8 in the article. Due to the sequence of the questions, it seems that the reviewer refers to table 6. For this reason, I take table 6 as the table object of the question. Secondly, it indicates that it is the antecedent of table 4. Table 4 shows the “Enhanced intermediate solution for OVE”. Since table 6 presents the cluster analysis for both OVE and ~OVE, I understand that the question could refer to presenting table 8 before table 3.

In relation to said question, the cluster analysis is performed after the identification of the solution because, as has stated in the work it analyzes whether the QCA solution formula obtained from the pooled data also can be found in each of the sub-populations in the data. Such sub-populations are the clusters that have been presented.

The need to first identify the intermediate solution and subsequently the cluster analysis is found in the fact that in the R SetMethods command, the solution to be used is incorporated as a value. With cluster analysis “we can set the diagnostic of how the solution holds throughout the different units and clusters by just imputing the solution [75, p. 527]

And by the way, you speak about 60 cities, but you do not have 60 of them in page 8.

I understand the reviewer is referring to the footnote. These notes begin with “Cities covered” because the solution is not able to explain all cities. What is presented are the cities explained by each solution.

“Coverage is the parameter that explains the empirical relevance of the solution. The coverage of the solution is fewer than the sum of the coverage’s of the different conjunctions that make up the solution, due to the existence of overlaps between said conjunctions. For this reason, unique coverage, the degree of coverage attributable to a single condition, is also presented [75]. For example, in the case of OVE, the coverage of the solution is 0.959, less than the sum of the coverage’s of the two conjunctions that compose it: SIM=0.917; INF*MAT*SEF=0.819. This is due, as indicated above, to the existence of overlapping cities. The cities explained by each solution are presented in the footnote. For example, in the case of OVE Oslo it is explained by both conjunctions”.

And please check this sentence in the abstract:  “The explanation of the readiness of cities to implement mobility innovations differs from the explanation of the negation of said readiness.” Seems to miss something – I would expect from.... to...

Thank you very much for the appreciation, the sentence has been modified.

“The explanation of the readiness of cities to implement mobility innovations is different to the explanation of the readiness negation”
